# Improving the Detection Sensitivity of a New Rapid Diagnostic Technology for Severe Acute Respiratory Syndrome Coronavirus 2 Using a Trace Amount of Saliva

**DOI:** 10.3390/diagnostics12112568

**Published:** 2022-10-22

**Authors:** Reiko Tokuyama-Toda, Chika Terada-Ito, Masaaki Muraoka, Toshikatsu Horiuchi, Tsuyoshi Amemiya, Airi Fukuoka, Yoshiki Hamada, Yusuke Takebe, Takashi Ogawa, Seiko Fujii, Toshihiro Kikuta, Shunsuke Sejima, Kazuhito Satomura

**Affiliations:** 1Department of Oral Medicine and Stomatology, School of Dental Medicine, Tsurumi University, 2-1-3, Tsurumi, Tsurumi-ku, Yokohama City 230-8501, Kanagawa, Japan; 2Certified Non-Profit Organization Biomedical Science Association, 2-20-8, Kamiosaki, Shinagawa-ku 141-0021, Tokyo, Japan; 3Department of Oral and Maxillofacial Surgery, Saiseikai Yokohamashi Tobu Hospital, 3-6-1, Shimosueyoshi, Tsurumi-ku, Yokohama City 230-8765, Kanagawa, Japan; 4Department of Oral and Maxillofacial Surgery, School of Dental Medicine, Tsurumi University, 2-1-3, Tsu-rumi, Tsurumi-ku, Yokohama City 230-8501, Kanagawa, Japan; 5Department of Oral and Maxillofacial Surgery, Tokyo Medical University Hachioji Medical Center, 1163, Tatemachi, Hachioji City 193-0998, Tokyo, Japan; 6Department of Oral and Maxillofacial Surgery, Shin-Yurigaoka General Hospital, 255, Furusawatsuko, Asao-ku, Kawasaki City 215-0026, Kanagawa, Japan

**Keywords:** SARS-CoV-2, COVID-19, diagnostic technology, mobile qPCR device, mouthwash, saliva

## Abstract

The early diagnosis and isolation of infected individuals with coronavirus disease 2019 (COVID-19) remain important. Although quantitative polymerase chain reaction (qPCR) testing is considered the most accurate test available for COVID-19 diagnosis, it has some limitations, such as the need for specialized laboratory technicians and a long turnaround time. Therefore, we have established and reported a rapid diagnostic method using a small amount of saliva as a sample using a lightweight mobile qPCR device. This study aimed to improve the existing method and increase the detection sensitivity and specificity. The detection specificity of CDC N1 and N2 was examined by improving qPCR reagents and polymerase chain reaction conditions for the previously reported method. Furthermore, the feasibility of detecting severe acute respiratory syndrome coronavirus 2 (SARS-CoV-2) viral RNA was examined using both the previous method and the improved method in patients with COVID-19. The results showed that the improved method increased the specificity and sensitivity. This improved method is useful for the rapid diagnosis of SARS-CoV-2.

## 1. Introduction

For coronavirus disease 2019 (COVID-19), which is currently a worldwide pandemic, the identification and quarantine of infected individuals is important for containment. There are various nucleic acid tests, including quantitative polymerase chain reaction (qPCR) [1,2,3,4,5,6], antigen tests [7,8,9,10,11,12], and antibody tests [13,14,15,16,17,18], for the diagnosis of severe acute respiratory syndrome coronavirus 2 (SARS-CoV-2) infection. qPCR for viral RNA detection is currently the most reliable and widely used method [1,2,3]. However, the conventional qPCR test has some drawbacks. For example, the turnaround time is long, a laboratory engineer with specialized knowledge/technique is needed, a relatively large benchtop device is required, and sample collection is complicated and involves a risk of infection to examiners [19]. Therefore, we developed and reported a new rapid diagnostic technology using a trace amount of saliva, using the PCR1100, which is a compact, lightweight, and portable qPCR device [20]. This method enabled us to diagnose patients in a short time of about 18 min on average and has the advantage of a much lower risk of infection due to the use og a mouthwash without any pretreatment of the sample. This rapid diagnostic method can contribute to the effective suppression of the spread of COVID-19. In this diagnostic procedure, N1 and N2 regions of the SARS-CoV-2 were targeted for quantitative gene amplification using specific primers and probes as recommended by the Centers for Disease Control and Prevention (CDC). Despite successfully amplifying the CDC N1 region during the diagnostic process in infected patients, the detection sensitivity of the N2 region was dramatically low, and CDC N1 was not detected in all cases [20]. Therefore, this study aimed to improve the detection sensitivity of CDC N2 and established qPCR conditions with higher sensitivity by optimizing annealing conditions and modifying polymerase chain reaction (PCR) reagents.

## 2. Materials and Methods

### 2.1. Samples

Samples were prepared as described in a previous study [20]. In brief, SARS-CoV-2 positive control RNA samples developed according to the National Institute of Infectious Diseases (NIID) protocol were provided by the NIID (for N primer probe Ver. 2 and N2 primer probe Ver. 2/Ver. 3). Following the CDC protocol, RNA samples were synthesized by FASMAC Co., Ltd. (Atsugi, Kanagawa, Japan) based on each primer sequence and information obtained from GenBank™ (accession number MN997409.1, position −10/+110 for 2019-nCoV_N1 and position +871/+980 for 2019-nCoV_N2). Ribonuclease P (RNase P) was also synthesized (accession number U77665.1, position +21/+130). Each positive control RNA was 10-fold serially diluted with 10 mM Tris-HCl (pH 8.0) including 10 µg/mL carrier RNA (RNA from baker’s yeast, Merck KGaA, Darmstadt, Germany).

### 2.2. Primers and Probes

Primers and probes were prepared as described in a previous study [20]. Based on the CDC 2020 protocol for SARS-CoV-2 [21], oligonucleotide primers and probes for quantitative reverse transcriptase PCR (qRT-PCR) were synthesized to target the N1 or N2 region by Nihon Gene Research Laboratories Inc. (Sendai, Japan) [20]. Based on the CDC protocol, oligonucleotide primers and probes were also synthesized to detect human RNase P.

### 2.3. RT-PCR

In this study, single-step qRT-PCR was performed for all tests using a PCR1100 device as described in a previous study [20]. In addition, the changes in this new version were as follows. A new version of the premix solution was prepared by modifying the primer concentration and magnesium chloride concentration in the original version of the premix solution. However, details cannot be disclosed due to intellectual property rights. In addition, the annealing/amplification temperature for CDC N2 was modified from 60 °C to 63 °C in the qRT-PCR conditions. Actually, after adding 3 µL of a sample (raw sample or positive control RNA) to the premix solution, the qRT-PCR was performed in a total reaction volume of 20 µL containing a final concentration of primer/probe for each target [20]. Following preliminary amplification reactions, the optimal concentration of RT Enzyme Mix was determined to be 0.25 µL for the CDC protocol.

The qRT-PCR conditions that were adequate for the PCR1100 device used in this study were programmed as follows: RT incubation and enzyme activation were serially performed at 50 °C for 150 s and at 95 °C for 15 s. Afterward, it was cycled 50 times at 95 °C for 3.5 s for denaturation and at 58 °C–63 °C for 8–16 s for the annealing/amplification for multichannel detection of SARS-CoV-2 and RNase P following the CDC protocol. Following preliminary amplification reactions, the optimal temperature for annealing/amplification was determined to be 63 °C for 8 s for the CDC protocol.

### 2.4. Detection of SARS-CoV-2 RNA in Mouthwash Obtained from Patients with COVID-19

In this study, two protocols were investigated: detection with the reagent and qPCR conditions used in the previous study [20] (Version 1) and detection with modified reagent and improved qPCR conditions (Version 2). We examined whether these two diagnostic protocols could detect CDC N1 and CDC N2 in 3 µL of mouthwash samples from patients with COVID-19 who had been diagnosed as SARS-CoV-2 positive by the conventional qPCR test. This study was approved by the Tsurumi University School of Dentistry Ethics Review Committee (No. 121002, 121014, 121018, 121027), the Saiseikai Yokohamashi Tobu Hospital Ethics Review Committee (No. 20210012), the Tokyo Medical University Hachioji Medical Center Ethics Review Committee (No. T2021-0230), and the Shin-Yurigaoka General Hospital Ethics Review Committee (No. 20220425-1). Of the patients admitted to Saiseikai Yokohamashi Tobu Hospital, Tokyo Medical University Hachioji Medical Center, and Shin-Yurigaoka General Hospital with a diagnosis of COVID-19, 6 adult patients agreed to participate in this study. Immediately after admission with a COVID-19 diagnosis based on conventional qPCR testing, an initial PCR test with the two protocols was performed.

## 3. Results

### 3.1. Analytical Limits of Detection (LoD)

Figure 1 shows the correlation between the concentration of synthetic RNA and the cycle threshold (Ct) value in the multichannel system of PCR1100. Under the old RT-PCR condition, Version 1, while a high linear correlation between the concentration of synthetic RNA and the Ct value was confirmed in each of the RNase P, SARS-CoV-2 CDC N1, and SARS-CoV-2 CDC N2, the LoD were 10 copies for RNase P and SARS-CoV-2 CDC N1, but 10,000 copies for SARS-CoV-2 CDC N2. This finding indicated that the detection sensitivity for N2 was 1000 times less than that for N1 and RNase P. In contrast, when using the new version, Version 2, while maintaining a high linear correlation between the concentration of synthetic RNA and the Ct value, LoD were 10 copies for RNase P, SARS-CoV-2 CDC N1, and SARS-CoV-2 CDC N2. These data revealed that multichannel detection using PCR1100 under the improved RT-PCR condition had a sufficiently high sensitivity for detecting the three targets. More importantly, this modification of the RT-PCR condition, which allows for the detection of small amounts of N2 RNA in addition to N1, may contribute to the improvement in the overall diagnostic specificity of this test.

### 3.2. Detection of CDC N1 and N2 of SARS-CoV-2 Viral RNA from Patients with COVID-19

In this study, we investigated whether CDC N1 and N2 of SARS-CoV-2 viral RNA in mouthwash from patients with COVID-19 who were diagnosed as SARS-CoV-2-positive by conventional qPCR testing by PCR1100 could be detected using both the old and new RT-PCR conditions. The samples were obtained from the mouthwash of six patients within 3 days from the onset. CDC N1 could be detected for all patients without any problem under both old and new conditions. In contrast, CDC N2 could be detected for all patients only under the new RT-PCR condition (Table 1). The case of patient 4 was presented as a typical case (Figure 2). CDC N1 of SARS-CoV-2 RNA was detected with a Ct value of 33.0, and CDC N2 was not detected using the old version. However, using the new version, both CDC N1 and N2 of SARS-CoV-2 RNA were detected, with Ct values of 32.0 and 34.0, respectively.

## 4. Discussion

With global uncertainty about the end of the pandemic in COVID-19, the containment of COVID-19 through early diagnosis and effective isolation of infected individuals remains important. Various diagnostic methods, including tests using point-of-care testing devices, have been reported [4,5,6,10,11,12,13,14,15,16,17,18,22]. Nonetheless, few tests have the same sensitivity and specificity as conventional qPCR tests. Although we also reported a rapid diagnostic method using a trace amount of saliva [20], there was a problem in the detection sensitivity of CDC N2. As previously reported [23,24,25], this low detection sensitivity for CDC N2 seems to be due to several factors, such as the propensity for primer dimer formation. Therefore, in this study, we modified the composition of the qRT-PCR premix solution and optimized the qPCR conditions to detect N2 and N1.

In vitro validation of LoD showed an improvement in the detection sensitivity of CDC N2 in the new qRT-PCR condition compared with the old one. Based on this result, qRT-PCR diagnosis using a PCR1100 device was performed under both the old and new conditions for patients with COVID-19 who were actually diagnosed as SARS-CoV-2-positive by the conventional qPCR test. As a result, CDC N2 was not detected in the previous version and was detected in all cases in the new version. Based on the fact that the final concentration of each primer increased, and the Ct value increased in the new version, which enabled the annealing temperature to be raised, the primers were considered to act more specifically without improper primer dimer formation, resulting in improved sensitivity and accurate detection of CDC N2. Regarding N1, we believe that there was no change in sensitivity or specificity because the annealing/amplification was not affected within the range of the conditions that changed in this study. We believe that these are due to the characteristics of the site at which the primers are set.

In this study, CDC N1 was detected in both the old and new versions, and CDC N2 was detected in the new version only when samples from patients within 3 days from the onset were used for diagnosis. However, it was difficult to detect N1 and N2 in both the new and old versions using samples from patients over 3 days from the onset. This result is inconsistent with our previous study [20], which showed that N1 was detected about 10 days after onset. This may be due to the difference in the timing of the two studies. This study was conducted during an outbreak of omicron strain, whereas the previous report was conducted during a delta outbreak. It has been reported that there are differences in the amount and period of virus excretion into saliva depending on the variant strain [26,27,28]. Still, what is important about this study is that CDC N2 was detected in all cases using the new version, indicating that the newer version successfully improved the detection sensitivity of CDC N2. The new version is now under further verification using a larger number of cases.

## 5. Conclusions

As demonstrated in previous study [20], this method has the advantages that it can be performed in a short time, is user-friendly and can easily be performed, even by medical staff who have no knowledge or experience in molecular cell biology, compared to conventional qPCR problems, such as cost, a long turnaround time, and securing human resources. In this study, the rapid diagnostic method for SARS-CoV-2 viral RNA using PCR1100 has been improved to detect CDC N1 and N2. This diagnostic method may be more suitable for screening tests because of its increased specificity and sensitivity, as well as its user-friendliness and promptness. To confirm the accuracy and usefulness of this diagnostic method and device, a larger number of cases should be examined by multiple medical institutions.

## Figures and Tables

**Figure 1 diagnostics-12-02568-f001:**
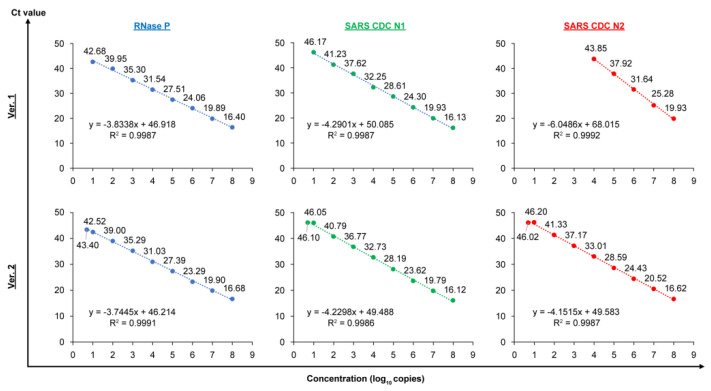
The correlation of each concentration of synthetic RNA and cycle threshold (Ct) value when the PCR1100 device was utilized to detect with multichannel of the old and new versions. Ver. 1 denotes the old version (qPCR condition: annealing/amplification temperature was 60 °C). Ver. 2 denotes the new version (qPCR condition: annealing/amplification temperature was 63 °C). The Ct value against each concentration is demonstrated by the number above each symbol. *x*: log_10_ copies; *y*: Ct value.

**Figure 2 diagnostics-12-02568-f002:**
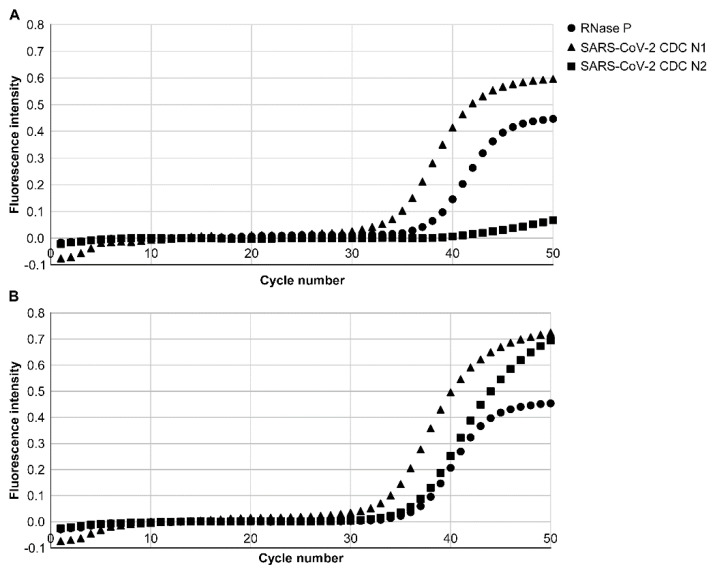
qPCR results by PCR1100 of patient 4. (**A**) Results of the qPCR using the old version (Ver. 1). RNase P was detected with a Ct value of 36.0, indicating that there was no problem with the PCR. CDC N1 increased at a Ct value of 33.0, and severe acute respiratory syndrome coronavirus 2 (SARS-CoV-2) viral RNA was detected. At this time, CDC N2 was not detected. (**B**) Results of the qPCR using the new version (Ver. 2). RNase P was detected with a Ct value of 35.0, indicating that there was no problem with the PCR. CDC N1 increased at a Ct value of 32.0, and CDC N2 increased at a Ct value of 34.0. Both CDC N1 and N2 of SARS-CoV-2 viral RNA were detected.

**Table 1 diagnostics-12-02568-t001:** Detection of severe acute respiratory syndrome coronavirus 2 (SARS-CoV-2) viral RNA (CDC Nl and N2) in patients with COVID-19 by PCR1100 using versions 1 and 2.

Patient	Age/Sex	Ver. 1 Ct Value			Ver. 2 Ct Value		
		RNase P	CDC N1	CDC N2	RNase P	CDC N1	CDC N2
1	90/M	35.3	39.0	(-)	33.3	37.4	43.2
2	56/F	37.8	37.0	(-)	34.1	33.9	46.3
3	50/M	31.1	45.5	(-)	29.7	39.9	46.1
4	25/F	36.0	33.0	(-)	35.0	32.0	34.0
5	70/M	33.5	30.0	(-)	31.3	28.7	38.3
6	90/M	32.0	35.2	(-)	30.3	31.4	35.6

RNase P: ribonuclease P; Ct: cycle threshold; M: male; F: female.

## Data Availability

The data that support the findings of this study are available from the corresponding author upon reasonable request.

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
