# Peer review of "Improving the Detection Sensitivity of a New Rapid Diagnostic Technology for Severe Acute Respiratory Syndrome Coronavirus 2 Using a Trace Amount of Saliva"

_diagnostics, 2022, doi:10.3390/diagnostics12112568_

Round 1

Reviewer 1 Report

The manuscript presents results from a study aimed to improve an existing method by improving the "detection sensitivity and specificity". Overall, it is very well written and clear, however, some details need clarification.

Major comment:

The main methodological innovation lies in "optimizing annealing conditions and modifying polymerase chain reaction". The current manuscript lacks sufficient details of how these are achieved. At multiple places, the gain in detection performance is attributed to "improved RT-PCR condition" but the specifics of the "RT-PCR condition" is not provided.

Figure 3 shows the main improvement of the new methods is in the detection of CDC N2. Why are the detection values very similar for other regions and only distinct for N2. This is not obvious? Why does the "optimized conditioning" only improve detection of N2?

The differences of the "conditions" from the "old" [ref 20] and "new" method is not clear. It is currently difficult to see what are the specific changes in condition from 'old' to 'new' method.

Some drawbacks of the conventional qPCR test based on qPCR are highlighted, e.g. requires specialized knowledge, etc. Please elaborate how the distinction and differences of the proposed methods (ref 20 and the current modification) with conventional qPCR overcomes these drawbacks.

Minor comment:

Line 178 "Tm" is used but not defined.

Author Response

Oct 5, 2022

Dear Reviewer #1

Diagnostics

Re:   Manuscript ID: diagnostics-1942801

Type of manuscript: Communication

Title: Improving the Detection Sensitivity of a New Rapid Diagnostic Technology for Severe Acute Respiratory Syndrome Coronavirus 2 Using a Trace Amount of Saliva

Thank you for your valuable comments concerning our manuscript entitled "Improving the Detection Sensitivity of a New Rapid Diagnostic Technology for Severe Acute Respiratory Syndrome Coronavirus 2 Using a Trace Amount of Saliva."

We have carefully studied your comments and made the necessary corrections, and are sending here the revised manuscript again. The corrected document has colored text.

Our responses to your comments are as follows:

Response to the comments of Reviewer #1

  1. The main methodological innovation lies in "optimizing annealing conditions and modifying polymerase chain reaction". The current manuscript lacks sufficient details of how these are achieved. At multiple places, the gain in detection performance is attributed to "improved RT-PCR condition" but the specifics of the "RT-PCR condition" is not provided.

Response

Thank you for your comment. I specified the PCR conditions in lines 88-89 as follows:

" In addition, the annealing/amplification temperature for CDC N2 was modified from 60°C to 63°C in the qRT-PCR conditions.” (L88 to L89 on page 2, Revision text)

  1. Figure 3 shows the main improvement of the new methods is in the detection of CDC N2. Why are the detection values very similar for other regions and only distinct for N2. This is not obvious? Why does the "optimized conditioning" only improve detection of N2?.

Response

Thank you for your comment.

Regarding N1, we believe that there was no change in sensitivity or specificity because the annealing/amplification was not affected within the range of the conditions changed in this study. On the other hand, N2 seems to have improved non-specific annealing/amplification as mentioned in the text. We believe that these are due to the characteristics of the site where the primers are set.

  1. The differences of the "conditions" from the "old" [ref 20] and "new" method is not clear. It is currently difficult to see what are the specific changes in condition from 'old' to 'new' method.

Response

Thank you for your comment. We clarified changes in method as follows:

“In this study, single-step qRT-PCR was performed for all tests using a PCR1100 device as described in a previous study [20]. In addition, the changes in this new version were as folloes. A new version of the premix solution was prepared by modifying the primer concentration and magnesium chloride concentration in the original version of the premix solution. However, details cannot be disclosed due to intellectual property rights. In addition, the annealing/amplification temperature for CDC N2 was modified from 60°C to 63°C in the qRT-PCR conditions.” (L83 to L89 on page 2, Revision text)

  1. Some drawbacks of the conventional qPCR test based on qPCR are highlighted, e.g. requires specialized knowledge, etc. Please elaborate how the distinction and differences of the proposed methods (ref 20 and the current modification) with conventional qPCR overcomes these drawbacks.

Response

Thank you for your comment. Based on comments from other reviewers, I added the following:

“As demonstrated in previous study [20], this method has the advantages that it can be performed in a short time, is user-friendly and can be performed easily even by medical staff who have no knowledge or experience in molecular cell biology, compared to conventional qPCR problems such as cost, long turnaround time, and securing human resources.” (L196 to L199 on page 6, Revision text)

  1. Line 178 "Tm" is used but not defined.

Response

Thank you for your comment. Tm is wrong, it should be Ct. It has changed.

We believe the manuscript has been improved satisfactorily and hope that it is now acceptable for publication in Diagnostics

Yours sincerely,

Reiko Tokuyama-Toda, DDS, PhD 

Reviewer 2 Report

Authors propose improving qPCR testing, considering the viral RNA of SARS-CoV-2, obtaining promising results which have been proved with some patients. Regards this work, the recommendations are listed as follows:

-      The results depicted in figure 1 do not show statistical parameters which can support the whole obtained results.

- Conclusions need more details

Author Response

Oct 5, 2022

Dear Reviewer #2

Diagnostics

Re:   Manuscript ID: diagnostics-1942801

Type of manuscript: Communication

Title: Improving the Detection Sensitivity of a New Rapid Diagnostic Technology for Severe Acute Respiratory Syndrome Coronavirus 2 Using a Trace Amount of Saliva

Thank you for your valuable comments concerning our manuscript entitled "Improving the Detection Sensitivity of a New Rapid Diagnostic Technology for Severe Acute Respiratory Syndrome Coronavirus 2 Using a Trace Amount of Saliva."

We have carefully studied your comments and made the necessary corrections, and are sending here the revised manuscript again. The corrected document has colored text.

Our responses to your comments are as follows:

Response to the comments of Reviewer #2

  1. The results depicted in figure 1 do not show statistical parameters which can support the whole obtained results.

Response

Thank you for your comment.

In this study, based on the results reported in the previous report [ref 20], the study aimed to improve the detection sensitivity of CDC N2. In Fig. 1, target quantification is guaranteed for both N1 and N2, which is sufficient for the purpose of Fig. 1. It has the same format as Fig. 2 of ref 20.

  1. Conclusions need more details

Response

Thank you for your comment. Based on comments from other reviewers, I added the following:

“As demonstrated in previous study [20], this method has the advantages that it can be performed in a short time, is user-friendly and can be performed easily even by medical staff who have no knowledge or experience in molecular cell biology, compared to conventional qPCR problems such as cost, long turnaround time, and securing human resources.” (L196 to L199 on page 6, Revision text)

We believe the manuscript has been improved satisfactorily and hope that it is now acceptable for publication in Diagnostics

Yours sincerely,

Reiko Tokuyama-Toda, DDS, PhD 

Round 2

Reviewer 1 Report

The authors have broadly addressed the major points in the revision. Some minor points to address in the revised version are:

- It will be very helpful to add the PCR conditions on each of the panels in Figure 1. This will help to clearly contrast the differences in these conditions and the corresponding changes in Ct values.

- The response by authors to Comment 2 in first round of review (copied below):

"Regarding N1, we believe that there was no change in sensitivity or specificity because the annealing/amplification was not affected within the range of the conditions changed in this study. On the other hand, N2 seems to have improved non-specific annealing/amplification as mentioned in the text. We believe that these are due to the characteristics of the site where the primers are set."

This point should be included in the manuscript to qualify the results and observation. Specifically, pertaining to the change in detection of N2. Moreover, as stated it is not clear as to why the N2 detection was improved but there are likely reasons which the authors have outlined and merit further investigation. Therefore, the authors should clearly articulate these limitations of the current study and include in the manuscript.

- Typo in line 85 (folloes).

Author Response

Oct 11, 2022

Dear Reviewer #1

Diagnostics

Re:   Manuscript ID: diagnostics-1942801

Type of manuscript: Communication

Title: Improving the Detection Sensitivity of a New Rapid Diagnostic Technology for Severe Acute Respiratory Syndrome Coronavirus 2 Using a Trace Amount of Saliva

Thank you for your valuable comments concerning our manuscript entitled "Improving the Detection Sensitivity of a New Rapid Diagnostic Technology for Severe Acute Respiratory Syndrome Coronavirus 2 Using a Trace Amount of Saliva."

We have carefully studied your comments again and made the necessary corrections, and are sending here the revised manuscript-2 again. The corrected document has colored text.

Our responses to your comments are as follows:

Response to the comments of Reviewer #1 (round 2)

  1. It will be very helpful to add the PCR conditions on each of the panels in Figure 1. This will help to clearly contrast the differences in these conditions and the corresponding changes in Ct values.

Response

Thank you for your comment.

I added the PCR conditions in figure 1.

  1. The response by authors to Comment 2 in first round of review (copied below):

"Regarding N1, we believe that there was no change in sensitivity or specificity because the annealing/amplification was not affected within the range of the conditions changed in this study. On the other hand, N2 seems to have improved non-specific annealing/amplification as mentioned in the text. We believe that these are due to the characteristics of the site where the primers are set."

This point should be included in the manuscript to qualify the results and observation. Specifically, pertaining to the change in detection of N2. Moreover, as stated it is not clear as to why the N2 detection was improved but there are likely reasons which the authors have outlined and merit further investigation. Therefore, the authors should clearly articulate these limitations of the current study and include in the manuscript.

Response

Thank you for your comment.

We added the following:

“Regarding N1, we believe that there was no change in sensitivity or specificity be-cause the annealing/amplification was not affected within the range of the condi-tions changed in this study. We believe that these are due to the characteristics of the site where the primers are set.” (L183 to L186 on page 5, Revision text-2)

  1. Typo in line 85 (folloes).

Response

Thank you for your comment. We changed. (L85, Revision text-2: “follows”)

We believe the manuscript has been improved satisfactorily and hope that it is now acceptable for publication in Diagnostics

Yours sincerely,

Reiko Tokuyama-Toda, DDS, PhD